# Likelihood Regret: An Out-of-Distribution Detection Score For Variational Auto-encoder

**Zhisheng Xiao** *
Computational and Applied Mathematics
University of Chicago
Chicago, IL, 60637
zxiao@uchicago.edu

**Qing Yan** *
Department of Statistics
University of Chicago
Chicago, IL, 60637
yanq@uchicago.edu

**Yali Amit**
Department of Statistics
University of Chicago
Chicago, IL, 60637
amit@marx.uchicago.edu

## Abstract

Deep probabilistic generative models enable modeling the likelihoods of very high dimensional data. An important application of generative modeling should be the ability to detect out-of-distribution (OOD) samples by setting a threshold on the likelihood. However, some recent studies show that probabilistic generative models can, in some cases, assign higher likelihoods on certain types of OOD samples, making the OOD detection rules based on likelihood threshold problematic. To address this issue, several OOD detection methods have been proposed for deep generative models. In this paper, we make the observation that many of these methods fail when applied to generative models based on Variational Auto-encoders (VAE). As an alternative, we propose Likelihood Regret, an efficient OOD score for VAEs. We benchmark our proposed method over existing approaches, and empirical results suggest that our method obtains the best overall OOD detection performances when applied to VAEs.

## 1 Introduction

In order to make reliable and safe decisions, deep learning models that are deployed for real life applications need to be able to identify whether the input data is anomalous or significantly different from the training data. Such data are called out-of-distribution (OOD) data. However, it is known that neural network classifiers can over-confidently classify OOD data into one of the training categories [37]. This observation poses a great challenge to the reliability and safety of AI [2], making OOD detection a problem of primary importance. Several approaches have been proposed to detect OOD data based on deep classifiers [17, 24, 13, 18]. Unfortunately, these methods cannot be applied to OOD detection for models trained without supervision, such as many generative models. An appealing OOD detection approach that may work for probabilistic generative models is to use their likelihood estimates. Such models can evaluate the likelihood of input data, and if a generative model fits the training data distribution well enough, it should assign high likelihood to samples from the training distribution and low likelihood to OOD samples.

Recent advances in deep probabilistic generative models [20, 46, 42, 21] make generative modeling of very high dimensional and complicated data such as natural images, sequences [38] and graphs [22]

---

possible. These models can evaluate the likelihood of input data easily and generate realistic samples, indicating that they succesfully approximate the distribution of training data. Therefore, it would appear promising to use deep generative models to detect OOD data [27]. However, some recent studies [34, 7] reveal a counter intuitive phenomenon that challenges the validity of unsupervised OOD detection using generative models. They observe that likelihoods obtained from current state-of-the-art deep probabilistic generative models fail to distinguish between training data and some obvious OOD input types that are easily recognizable by humans. For example, [34] shows that generative models trained on CIFAR-10 output higher likelihood on SVHN than on CIFAR-10 itself, despite the fact that images in CIFAR-10 (contains dogs, trucks, horses, etc.) and SVHN (contains house numbers) have very different semantic content.

At this point, no effective method has been discovered to ensure these generative models make the correct likelihood assignment on OOD data. Alternatively, some new scores based on likelihood are proposed to alleviate this issue [40, 35, 43]. The OOD detection is performed by setting thresholds on the new scores rather than on likelihood. Some of these methods obtain impressive OOD detection performance on invertible flow-based models [21] and auto-regressive models [42]. Interestingly, we observe that these scores can be much less effective for Variational Auto-encoders (VAE), an important type of probabilistic generative models. The failure of current OOD scores on VAE suggests that a new score is necessary. To this end, in this paper we propose a simple yet effective metric called Likelihood Regret (LR) to detect OOD samples with VAEs. The Likelihood Regret of a single input can be interpreted as the log ratio between its likelihood obtained by the posterior distribution optimized individually for that input and the likelihood approximated by the VAE. We conduct comprehensive experiments to evaluate our proposed score on a variety of image OOD detection tasks, and we show that it obtains the best overall performance.

## 2 Background

### 2.1 Variational Auto-Encoder

VAE [20, 41] is an important type of deep probabilistic generative model with many practical applications [28, 30, 16]. It uses a latent variable $\mathbf{z}$ with prior $p(\mathbf{z})$, and a conditional distribution $p_\theta(\mathbf{x}|\mathbf{z})$, to model the observed variable $\mathbf{x}$. The generative model, denoted by $p_\theta(\mathbf{x})$, can be formulated as $p_\theta(\mathbf{x}) = \int_{\mathcal{Z}} p_\theta(\mathbf{x}|\mathbf{z})p(\mathbf{z})\mathrm{d}\mathbf{z}$. However, direct computation of this likelihood is intractable in high dimensions, so variational inference is used to derive a lower bound on the log likelihood of $\mathbf{x}$. This leads to the famous evidence lower bound (ELBO):

$$\log p_\theta(\mathbf{x}) \geq \mathbb{E}_{q_\phi(\mathbf{z}|\mathbf{x})}\left[\log p_\theta(\mathbf{x}|\mathbf{z})\right] - D_{\mathrm{KL}}\left[q_\phi(\mathbf{z}|\mathbf{x})\|p(\mathbf{z})\right]$$
$$\triangleq \mathcal{L}(\mathbf{x};\theta,\phi), \tag{1}$$

where $q_\phi(\mathbf{z}|\mathbf{x})$ is the variational approximation to the true posterior distribution $p_\theta(\mathbf{z}|\mathbf{x})$. Both $q_\phi(\mathbf{z}|\mathbf{x})$ and $p_\theta(\mathbf{x}|\mathbf{z})$ are parameterized by neural networks with parameters $\phi$ (encoder) and $\theta$ (decoder), respectively. The VAE is trained by maximizing $\mathcal{L}(\mathbf{x};\theta,\phi)$ over the training data.

Unlike generative models using exact inference so that the likelihood can be directly computed, VAE only outputs a lower bound of the log likelihood and the exact log likelihood needs to be estimated, usually by an importance weighted lower bound [5]:

$$\log p_\theta(\mathbf{x}) \geq \mathbb{E}_{\mathbf{z}^1,\dots,\mathbf{z}^K \sim q_\phi(\mathbf{z}|\mathbf{x})}\left[\log \frac{1}{K}\sum_{k=1}^{K} \frac{p_\theta\left(\mathbf{x}|\mathbf{z}^k\right)p(\mathbf{z}^k)}{q_\phi\left(\mathbf{z}^k|\mathbf{x}\right)}\right] \triangleq \mathcal{L}_K(\mathbf{x};\theta,\phi), \tag{2}$$

where each $\mathbf{z}^k$ is a sample from the variational posterior $q_\phi\left(\mathbf{z}|\mathbf{x}\right)$.

While the prior $p(\mathbf{z})$ and the variational posterior $q_\phi\left(\mathbf{z}|\mathbf{x}\right)$ are often chosen to be Gaussians, there are multiple choices for the decoding distribution $p_\theta\left(\mathbf{x}|\mathbf{z}\right)$ depending on the type of data. In this paper, we follow the settings of VAE experiments in [34] and choose the decoding distribution to be a factorial 256-way categorical distribution (corresponding to 8-bit image data) on each pixel. Note that the same data distribution is assumed by PixelCNN [46].

### 2.2 Problems with OOD Detection using Probabilistic Generative Models

Suppose we have a set of $N$ training samples $\{\mathbf{x}_i\}_{i=1}^{N}$ drawn from some underlying data distribution $\mathbf{x}_i \sim p(\mathbf{x})$. Our goal is to decide whether a test sample $\mathbf{x}$ is OOD, which, by definition in [29], means

that $\mathbf{x}$ has low density under $p(\mathbf{x})$. Probabilistic generative models $p_\theta(\mathbf{x})$ are trained on the set of training samples by maximizing the likelihood (or lower bound of likelihood). It is well known that maximizing the likelihood $p_\theta(\mathbf{x})$ is equivalent to minimizing $D_{\mathrm{KL}}\left[p(\mathbf{x})\|p_\theta(\mathbf{x})\right]$, and thus a well trained generative model provides a good approximation to the true data distribution $p(\mathbf{x})$. Therefore, OOD data should have low likelihood under $p_\theta(\mathbf{x})$, since they stay in low probability regions of $p(\mathbf{x})$.

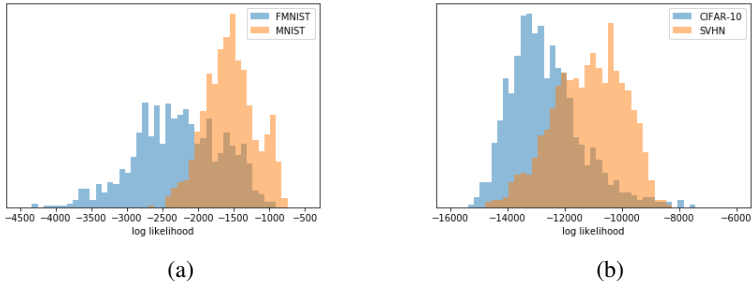

(a)  (b)

Figure 1: Histogram that compares the log likelihood of test samples from **(a)**: Fsahion MNIST and MNIST on a VAE trained on Fashion MNIST, and **(b)**: CIFAR-10 and SVHN on a VAE trained on CIFAR-10. Both experiments show that VAEs may assign high likelihoods to OOD samples.

However, the above argument fails in practice, as noted in [34, 7]. In particular, [34] observe that almost all major types of probabilistic generative models, including VAE, flow-based model and auto-regressive model, can assign spuriously high likelihood to OOD samples. In Figure 1, we confirm that such a likelihood misalignment does exist on VAE, which is the model we focus on in this paper. We further observe that VAEs obtain surprisingly good reconstruction quality on OOD data (Figure 2), indicating that they model OOD samples very well. These observations suggest that VAEs do not really regard some samples from completely different distributions as OOD, and it is extremely unreliable to use likelihood as an OOD detector.

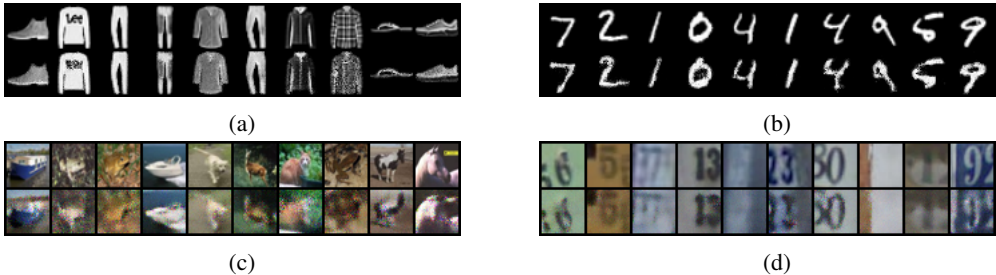

Figure 2: For each subfigure, the top row contains original images and the bottom row contains the reconstructed images. **(a), (b):** reconstruction of Fashion MNIST and MNIST images by a VAE trained on Fashion MNIST. **(c), (d):** reconstruction of CIFAR-10 and SVHN images by a VAE trained on CIFAR-10.

## 3  Related Work

Some specific types of generative models are shown to correctly assign low likelihoods to OOD samples. For example, [15] show that energy-based models do not suffer seriously from the likelihood misalignment issue. However, the test likelihoods of in-distribution and OOD samples still exhibit significant overlap. [32] shows that bidirectional-inference VAEs with very deep hierarchy of latent variables can slightly alleviate the likelihood misalignment issue, however, this comes at the cost of much worse likelihood estimates.

To the best of our knowledge, there is no consistent way to train generative models that can effectively detect OOD samples only by looking at likelihood, and therefore people seek to design new OOD scores. [7] observe that OOD samples have higher variance likelihood estimates under different independently trained models. Although the metric obtained from an ensemble of models performs well, training multiple models can be computationally expensive. [35] use an explicit test of typicality

|          | VAE  | Glow | PixelCNN |          | VAE  | Glow | PixelCNN |
|----------|------|------|----------|----------|------|------|----------|
| FMNIST   | 3.20 | 3.25 | 2.68     | CIFAR-10 | 4.12 | 4.05 | 3.57     |
| MNIST    | 2.18 | 2.10 | 1.51     | SVHN     | 3.63 | 2.65 | 2.39     |
| Constant | 4.21 | 1.35 | 4.01     | Constant | 2.49 | 0.76 | 0.85     |
| Noise    | 8.89 | 8.96 | 7.71     | Noise    | 5.98 | 9.38 | 10.17    |

(a) Trained on Fashion MNIST                    (b) Trained on CIFAR-10

Table 1: Average BPD of different test datasets for VAE, Glow and PixelCNN trained on Fashion MNIST and CIFAR-10.

and [45] propose an OOD detection score that leverages the batch normalization. However, both [35, 45] can only determine whether a *batch* of samples is an outlier or not, which greatly limits their applications to the real OOD detection task, where normally we want to detect if a single sample is in- or out-of-distribution.

Perhaps [40, 43] have the closest connection with our work. [40] propose the use of a likelihood-ratio test by taking the ratio between the likelihood obtained from the model and from a background model which is trained on random perturbations of input data. [43] hypothesizes that the likelihood of generative models are biased by the complexity of the inputs, and they offset the bias by a factor that measures the input complexity. They use the length of lossless compression of the image as the complexity factor, and their OOD score can also be interpreted as a likelihood-ratio test statistic by regarding the compressor as a universal model. [40, 43] obtain great OOD detection performances with Glow and Pixel-CNN, however, neither of them evaluates their methods on VAE. Later in this paper, we will show that their OOD scores do not work well with VAEs, suggesting the need to design an effective OOD score for VAEs.

Previously, Auto-encoders and VAEs were widely used for anomaly detection [3], and empirical success was achieved in web applications [49] and time series [51]. However, their main ideas are based on the hope that VAEs cannot reconstruct OOD samples well [1, 52, 11], which was later proven to be false in many cases. [12] incorporates both reconstruction loss and the Mahalanobis distance [26] in the latent space as an OOD detection score. Their method improves the performance of reconstruction based OOD detection, but we will show that it is still not effective in many experiments. [9] propose a way to enable OOD detection with VAE, but they require training with negative samples.

## 4   Likelihood Regret for OOD Detection using VAEs

### 4.1   Why We Need a New OOD Score for VAE?

Before introducing our method, we would like to emphasize why it is necessary to design a metric of OOD detection for VAE. One might ask, given that OOD scores like [40, 43] work so well for Glow and PixelCNN, why not just apply them to VAEs? We point out a key difference between VAE and other generative models in Table 1, where we trained different generative models on Fashion MNIST and CIFAR-10 and report the test bits-per-dimension (BPD) of different datasets. The BPD is computed by normalizing the negative log likelihood by the dimension of an input: $\text{BPD}(\mathbf{x}) = \frac{-\log p_\theta(\mathbf{x})}{\log(2) \cdot d}$. We observe that while all generative models exhibit similar behavior of assigning high likelihoods to certain types of OOD samples, the relative change in average likelihood across different datasets are different. In particular, the average test likelihoods of VAE across different datasets have a much smaller range than that of Glow and PixelCNN, suggesting that the likelihoods of in-distribution and OOD samples are much less "separated away" in VAE. The reason is probably that flow-based and auto-regressive models try to model each pixel of the input image, while the bottleneck structure in VAE forces the model to ignore some information.

Less separated likelihoods of in-distribution and OOD samples make OOD detection for VAE harder, as some OOD scores rely on the gap of likelihoods. We will empirically show in Section 5 that current state-of-the-art generative model OOD scores are much less effective for VAE, partly due to the smaller gap of likelihoods. This suggests the need for a new OOD score for VAE.

---

**Algorithm 1** Computing Likelihood Regret (LR)

---

**Input:** Test sample $\mathbf{x}$, trained VAE $(E_{\phi^*}, D_{\theta^*})$, number of posterior samples for likelihood estimation $K$, number of optimization step $S$, learning rate $\gamma$.

1: $L_{\text{VAE}} = \mathcal{L}_K(\mathbf{x}; \theta^*, \phi^*)$       ▷ Estimate the log likelihood of $\mathbf{x}$ under the VAE model by (2)
2: Set $\phi = \phi^*$
3: **for** $S$ iterations **do**
4:      $\phi \leftarrow \text{Adam}(\phi, \nabla_\phi(-\mathcal{L}(\mathbf{x}; \theta^*, \phi)), \gamma)$      ▷ Optimize $\phi$ by maximizing the ELBO objective
5: $L_{\text{OPT}} = \mathcal{L}_K(\mathbf{x}; \theta^*, \phi)$       ▷ Estimate the log likelihood of $\mathbf{x}$ with optimized encoder
6: $\text{LR} = L_{\text{OPT}} - L_{\text{VAE}}$

---

## 4.2 Our Proposed Score

Our proposed OOD detection score, called Likelihood Regret (LR), measures the log likelihood improvement of the model configuration that maximizes the likelihood of an individual sample over the configuration that maximizes population likelihood. Intuitively, if a generative model is well trained on the training data distribution, given an in-distribution test sample, the improvement of likelihood by replacing the current model configuration with the optimal one for the single sample should be relatively small, hence resulting in low LR. In contrast, for an OOD test sample, since the model has not seen similar samples during training, the current model configuration is much less likely to be close to the optimal one, hence the LR could be large. Therefore, LR can serve as a good OOD detection score. However, in practice, a generative model can easily overfit when being optimized on a single sample and obtain very high likelihood, thus we have to seek some form of regularization on the model configuration that is being optimized. Luckily, the bottleneck structure of VAE provides a natural regularization, by restricting the optimization to the parameters of the variational posterior distribution.

Formally, suppose we have a VAE with encoder $E_\phi$ and decoder $D_\theta$. As commonly used in VAE [20], $q_\phi(\mathbf{z}|\mathbf{x}) \sim \mathcal{N}(\mu_\mathbf{x}, \sigma_\mathbf{x}^2 \mathbf{I})$, so the encoder outputs the mean and variance: $E_\phi(\mathbf{x}) = (\mu_\mathbf{x}, \sigma_\mathbf{x})$. For clarity, we use $\tau(\mathbf{x}, \phi)$ to denote the sufficient statistics $(\mu_\mathbf{x}, \sigma_\mathbf{x})$ of $q_\phi(\mathbf{z}|\mathbf{x})$. Further we express ELBO in (1) as $\mathcal{L}(\mathbf{x}; \theta, \tau(\mathbf{x}, \phi))$ to emphasize its direct dependency on $\tau(\mathbf{x}, \phi)$. During training, based on empirical risk minimization (ERM) criterion, our objective is to obtain network parameters $\Theta^* = (\phi^*, \theta^*)$ that maximize the population log likelihood $\log p_\theta(\mathbf{x})$. In other words, $\Theta^*$ are the parameters that achieve best *average* log likelihood over the finite training set. In practice, we train VAE by maximizing ELBO instead of log likelihood. Since ELBO is a lower bound of log likelihood, we can regard maximizing EBLO as a good surrogate for maximizing log likelihood, so

$$(\phi^*, \theta^*) \approx \arg\max_{(\phi, \theta)} \frac{1}{n} \sum_{i=1}^{n} \mathcal{L}(\mathbf{x}_i; \theta, \tau(\mathbf{x}_i, \phi)). \tag{3}$$

Since $q_\phi(\mathbf{z}|\mathbf{x})$ can be fully determined by $\tau(\mathbf{x}, \phi)$, we also denote $\Theta = (\tau, \theta)$ or $\Theta = (\tau(\cdot, \phi), \theta)$ as an abuse of notation.

For a specific test input $\mathbf{x}$, we can fix the decoder parameters $\theta^*$, and find the optimal configuration of the variational posterior distribution parameter $\hat{\tau}(\mathbf{x}) = (\hat{\mu}_\mathbf{x}, \hat{\sigma}_\mathbf{x})$ that maximizes its *individual* ELBO:

$$\hat{\tau}(\mathbf{x}) = \underset{\tau}{\text{argmax}} \, \mathcal{L}(\mathbf{x}; \theta^*, \tau). \tag{4}$$

In other words, $\hat{\tau}(\mathbf{x})$ is the optimal posterior distribution of the latent variable $\mathbf{z}$ given the particular input $\mathbf{x}$ and the optimal decoder $\theta^*$ obtained from the training set. Now we define the Likelihood Regret (LR) of the input data $\mathbf{x}$ as

$$\text{LR}(\mathbf{x}) = \mathcal{L}(\mathbf{x}; \theta^*, \hat{\tau}(\mathbf{x})) - \mathcal{L}(\mathbf{x}; \theta^*, \phi^*). \tag{5}$$

LR can also be interpreted as the log ratio of the likelihood obtained from the generative model (VAE) with the optimal configuration of the variational posterior distribution for an individual input, to its likelihood obtained from the VAE trained on training set. This interpretation connects our method to [40, 43], which also have a likelihood ratio interpretation. The naming of our method is motivated by the concept of regret in online learning, which measures how 'sorry' the learner is, in retrospect, not to have followed the predictions of the optimal hypothesis [44].

**Implementation:** The major component of evaluating LR is computing $\hat{\tau}(\mathbf{x})$ defined in (4), namely the configuration $\tau(\mathbf{x})$ that maximizes the likelihood (ELBO) for a single sample. To do this, we fix the decoder $\theta^*$, and apply iterative optimization algorithms on $\tau$ with initialization $\tau^*(\mathbf{x}) = E_{\phi^*}(\mathbf{x})$ using $\mathcal{L}(\mathbf{x}; \theta^*, \tau)$ as the objective function until convergence. As an alternative approach, instead of optimizing $\tau$ directly, we can also optimize the parameters $\phi$ (initialized as $\phi^*$) of the encoder given $\mathbf{x}$ and $\theta = \theta^*$. We summarize the computation of Likelihood Regret in Algorithm 1.

## 5 Results

In this section, we conduct experiments to benchmark the performance of Likelihood Regret on OOD detection tasks on image datasets. For most experiments, we train VAEs with samples only from the training set of in-distribution data (Fashion-MNIST and CIFAR-10), and use test samples from different datasets to measure the OOD performances. Details regarding datasets and experimental settings can be found in Appendix A.

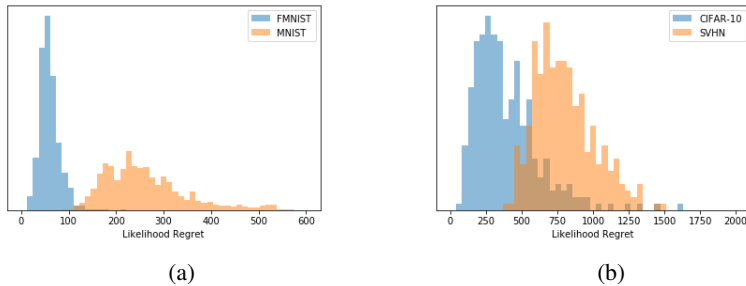

|     (a)     |     (b)     |

Figure 3: Histogram that compares the Likelihood Regret of test samples from **(a)**: Fsahion MNIST and MNIST on a VAE trained on Fashion MNIST, and **(b)**: CIFAR-10 and SVHN on a VAE trained on CIFAR-10. Both experiments show that OOD samples tend to have higher LR, as expected.

We first follow the setting of [34] and conduct the following two experiments as a proof-of-concept: (a) Fashion-MNIST as in-distribution and MNIST as OOD, and (b) CIFAR-10 as in-distribution and SVHN as OOD. We train VAEs on the training set of Fashion MNIST and CIFAR-10, and compute the Likelihood Regret for 1000 random samples from the test set of in-distribution data and corresponding OOD data. We plot the histograms of LR in Figure 3. Comparing Figure 1 with Figure 3, we observe that the VAE will assign higher likelihoods to OOD samples, while Likelihood Regret can largely correct such likelihood misalignment. OOD samples typically have larger LR than in-distribution samples, which confirms the effectiveness of our OOD score.

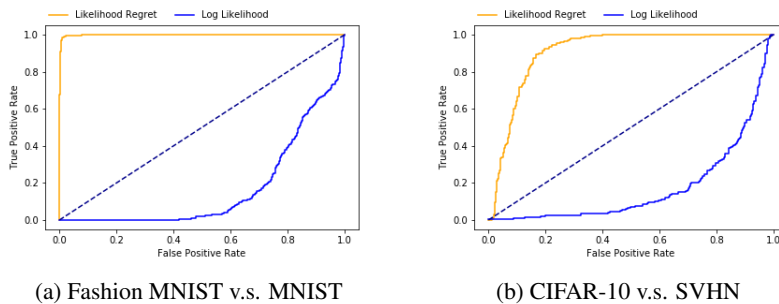

| (a) Fashion MNIST v.s. MNIST | (b) CIFAR-10 v.s. SVHN |

Figure 4: Comparing the ROC curves of using Likelihood Regret and Log Likelihood for OOD detection. On Fashion MNIST v.s. MNIST experiment, Likelihood Regret improves the AUC-ROC of OOD detection from 0.165 to 0.999. On CIFAR-10 v.s. SVHN experiment, Likelihood Regret improves the AUC-ROC of OOD detection from 0.161 to 0.876.

### 5.1 Metrics

Like other OOD scores, Likelihood Regret needs a threshold when applied to the OOD detection tasks. The choice of a threshold depends on the particular application. To quantitatively evaluate our method, we mainly use Area Under the Curve-Receiver Operating Characteristics (AUCROC↑)

as a metric. AUCROC is commonly used in OOD detection tasks [17], and it provides an effective performance summary across different score thresholds [14, 33]. In Figure 4, we plot the ROC curves for the above two experiments, and results show that log likelihood is an extremely bad OOD detector as it obtains AUC-ROC much lower than 0.5 (random guessing), while LR achieves good OOD performances. In addition to AUCROC, we also include Area Under the Curve-Precision Recall Curve (AUCPRC↑) and the False Positive Rate at 80% True Positive Rate (FPR80↓) as quantitative measurements.

## 5.2 Quantitative Comparison with Other OOD Scores

After a simple verification of the effectiveness of our proposed OOD score, we carefully study its performances and compare it with competing methods. In our comparison we use likelihood as a simple baseline, and include OOD scores discussed in Section 3, including two variants of input complexity adjusted score (**IC (png)** and **IC (jp2)**) [43], likelihood ratio with background model (**Likelihood Ratio**) [40] as well as latent Mahalanobis distance (**LMD**) [4]. Details of these methods and their implementations can be found in Appendix A.3. We present two implementations of Likelihood Regret: one optimizes the whole encoder (**LR$_\mathbf{E}$**), and the other only optimizes $(\mu_\mathbf{x}, \sigma_\mathbf{x})$ (**LR$_\mathbf{Z}$**).

|  | LR$_E$ | LR$_Z$ | Likelihood | IC (png) | IC (jp2) | Likelihood Ratio | LMD |
|---|---|---|---|---|---|---|---|
| MNIST | **0.988** | 0.967 | 0.201 | 0.946 | 0.553 | 0.924 | 0.877 |
| CIFAR-10 | 0.997 | 0.998 | 1 | 0.907 | 0.999 | 0.968 | 0.995 |
| SVHN | 1 | 1 | 0.999 | 0.992 | 1 | 0.785 | 0.995 |
| KMNIST | **0.994** | 0.983 | 0.731 | 0.708 | 0.599 | 0.983 | 0.955 |
| NotMNIST | 0.999 | 1 | 0.943 | 0.923 | 0.966 | 0.996 | 0.998 |
| Noise | 1 | 0.963 | 1 | 0.453 | 1 | 1 | 1 |
| Constant | 1 | 1 | 0.928 | 1 | 1 | 0.775 | 0.981 |

(a) VAE trained on Fashion MNIST

|  | LR$_E$ | LR$_Z$ | Likelihood | IC (png) | IC (jp2) | Likelihood Ratio | LMD |
|---|---|---|---|---|---|---|---|
| MNIST | **0.998** | 0.976 | 0.008 | 0.994 | 0.988 | 0.792 | 0.027 |
| FMNIST | 0.991 | 0.963 | 0.074 | **0.992** | 0.990 | 0.807 | 0.183 |
| SVHN | 0.875 | 0.843 | 0.193 | **0.912** | 0.908 | 0.265 | 0.279 |
| LSUN | **0.691** | 0.640 | 0.494 | 0.624 | 0.315 | 0.632 | 0.527 |
| CelebA | **0.714** | 0.690 | 0.465 | 0.641 | 0.564 | 0.447 | 0.576 |
| Noise | 0.994 | 0.922 | 1 | 0.032 | 0.054 | 1 | 0.983 |
| Constant | 0.974 | 0.924 | 0.258 | 1 | 1 | 0.470 | 0.431 |

(b) VAE trained on CIFAR-10

Table 2: AUCROC of Likelihood Regret (LR) and other OOD detection scores on different datasets. Each row contains the results of an OOD dataset.

The main results of this work are presented in Table 2, which shows the AUCROC scores of different methods on different datasets. We also present results of AUPRC and FPR80 in Table 6 and Table 7 in Appendix B. We see that OOD detection scores exhibit largely consistent behavior across these metrics. We make several important observations from these results:

1. We confirm that likelihood is problematic in OOD detection. For example, VAE trained on CIFAR-10 not only assigns significantly higher test likelihood on SVHN, but also on MNIST, Fashion MNIST and random constant images. The severe likelihood misalignment is reflected in an AUCROC value close to 0.

2. Likelihood Regret successfully corrects the misalignment of likelihood and obtains good OOD detection performances across all tasks, as the AUCROC values are close to the optimal value 1 on all experiments excepts for CIFAR-10 v.s CelebA and CIFAR-10 v.s LSUN. Note that these are the hard cases as all methods do not perform well. Samples from these datasets have similar texture as samples from CIFAR-10, and therefore it is hard for generative models trained without class labels to distinguish between them. Further, we observe that optimizing the encoder leads to slightly better performance than optimizing

|          | LR$_\text{ED}$ | LR$_\text{D}$ |          | LR$_\text{ED}$ | LR$_\text{D}$ |
|----------|------|------|----------|------|------|
| MNIST    | 0.124 | 0.189 | SVHN     | 0.195 | 0.212 |
| KMNIST   | 0.495 | 0.609 | LSUN     | 0.468 | 0.526 |
| NotMNIST | 0.969 | 0.947 | CelebA   | 0.488 | 0.655 |
| Noise    | 0.001 | 0.004 | Noise    | 0.002 | 0     |
| Constant | 0.989 | 1     | Constant | 0.322 | 0.435 |

(a) VAE trained on Fashion MNIST          (b) VAE trained on CIFAR-10

Table 3: AUCROC of Likelihood Regret (LR) obtained by optimizing different components of the VAE. LR$_\text{ED}$ corresponds to optimizing both the encoder and decoder, and LR$_\text{D}$ corresponds to optimizing the decoder only.

$(\mu_\mathbf{x}, \sigma_\mathbf{x})$ only. One possible explanation is that, for in distribution samples, optimizing the encoder is more constrained than directly optimizing $z$, which prevents the latent variables from moving too much.

3. While competing methods also obtain good OOD detection performance on some tasks, all of them exhibit severe issues on certain specific tasks. For example, likelihood ratio with background model fails on the classic task of detecting SVHN from in-distribution CIFAR-10 (AUCROC only 0.28); both variants of input complexity adjusted likelihood fail almost completely on distinguishing between random uniform noise and CIFAR-10 (AUCROC close to 0); latent Mahalanobis distance does not perform well on almost all OOD datasets when CIFAR-10 is the in-distribution dataset. These failure cases suggest that none of these OOD scores can be safely applied, at least for VAE models. In contrast, likelihood regret is shown to be effective on all the tasks.

Overall, based on results from Table 1, 6 and 7, we claim that LR achieves the best OOD detection performance. On those tasks where the performance of LR is not the best it is still very close to the best results. More importantly, it achieves good performance without any failure case, while all competing methods are shown to be ineffective on some experiments. Interestingly, the failure cases of competing methods do not exist in their papers where the scores are computed for Glow and PixelCNN. This is partly explained in Section 4.1. For example, on average, the difference of BPD returned by VAE is only 1.86 nats between CIFAR-10 and noise images, while for Glow and PixelCNN, the differences are 5.33 nats and 6.6 nats, respectively. However, the noise images have much larger complexity measurement than CIFAR-10, and the gap of complexity measurement will override the gap of likelihood, leading the input complexity adjusted score to make the wrong decision. Indeed, we observe that this happens sometimes, even for flow based models (see Appendix D). As for likelihood ratio with background model, this score heavily relies on the contrast between how well a single pixel is modeled by the main model and the background model. However, VAEs are not designed for modeling each single pixel, and the bottleneck structure will "smooth out" the background, making the contrast with a background model much less effective.

In summary, the experiments provide strong evidence that current state-of-the-art OOD scores for generative models may not be applicable to VAE, while our proposed score achieves good OOD detection results on a variety of tasks. To the best of our knowledge, Likelihood Regret is the only effective OOD score that can correct the likelihood misalignment of VAE.

**Optimizing other components of the VAE:** One key reason for the effectiveness of LR on VAE is that we only optimize the configuration of latent variables for a single test input, which is done by optimizing $\mathbf{z}$ directly or optimizing the encoder parameters. The bottleneck structure at the latent variable provides natural regularization that avoids overfitting on the single example. We argue that optimizing other components of the VAE will be less effective, as the optimization can easily overfit any test sample regardless if it is OOD or not. To show this, we perform ablation studies that optimize either the decoder only or the whole VAE (both encoder and decoder), and results are shown in Table 3. We observe that these alternative approaches are significantly worse than optimizing the latent variables or encoder parameters.

**Results on $\beta-$VAE:** VAEs are typically trained with the ELBO objective in (1), however, in practice, sometimes we train $\beta-$VAEs, where there is a coefficient $\beta \neq 1$ on the KL divergence term. Usually we use $\beta < 1$ for better sample quality [10, 48], and $\beta > 1$ for disentanglement in latent space [19, 6].

| | $\beta = 0.1$ | | $\beta = 0.5$ | | $\beta = 5$ | | $\beta = 10$ | |
|---|---|---|---|---|---|---|---|---|
| | Likelihood | $LR_E$ | Likelihood | $LR_E$ | Likelihood | $LR_E$ | Likelihood | $LR_E$ |
| MNIST | 0.191 | 0.988 | 0.154 | 0.948 | 0.231 | 0.997 | 0.225 | 0.966 |
| KMNIST | 0.704 | 0.997 | 0.679 | 0.985 | 0.696 | 0.996 | 0.698 | 0.952 |
| NotMNIST | 0.986 | 1 | 0.983 | 1 | 0.985 | 1 | 0.99 | 0.999 |
| Noise | 1 | 0.989 | 1 | 0.992 | 1 | 0.996 | 1 | 0.991 |
| Constant | 0.982 | 0.995 | 0.964 | 0.998 | 0.928 | 0.998 | 0.968 | 0.986 |

(a) $\beta-$VAEs trained on Fashion MNIST

| | $\beta = 0.1$ | | $\beta = 0.5$ | | $\beta = 5$ | | $\beta = 10$ | |
|---|---|---|---|---|---|---|---|---|
| | Likelihood | $LR_E$ | Likelihood | $LR_E$ | Likelihood | $LR_E$ | Likelihood | $LR_E$ |
| SVHN | 0.189 | 0.847 | 0.171 | 0.835 | 0.194 | 0.866 | 0.153 | 0.873 |
| LSUN | 0.474 | 0.645 | 0.465 | 0.655 | 0.469 | 0.654 | 0.442 | 0.657 |
| CelebA | 0.499 | 0.709 | 0.477 | 0.714 | 0.528 | 0.718 | 0.462 | 0.706 |
| Noise | 1 | 1 | 1 | 0.999 | 1 | 0.994 | 1 | 0.9735 |
| Constant | 0.319 | 0.964 | 0.279 | 0.947 | 0.268 | 0.962 | 0.284 | 0.954 |

(b) $\beta-$VAEs trained on CIFAR-10

Table 4: AUCROC of Likelihood Regret (LR) and Likelihoods for $\beta-$VAEs on different datasets.

We evaluate the OOD detection performances of LR on $\beta-$VAE variants in Table 4. From Table 4, we observe that LR behaves consistently across different values of $\beta$.

**On the capacity of the VAEs:** One concern regarding the effectiveness of Likelihood Regret is that it may depend on the capacity of VAEs. For example, a VAE with large capacity is easier to optimize for the encoder configuration on a single sample, leading to larger Likelihood Regret score. we evaluate LR on VAEs with different capacities, and results are presented in Appendix G, Table 9. We conclude that the performance of LR is robust to the capacity of VAEs. The AUCROC slightly drops for VAEs with large capacity trained on CIFAR-10. By inspection, we observe that large VAEs overfit the training data, as the test NLL on CIFAR-10 is even larger than that of the baseline VAE. In the case of overfitting, the LR for in-distribution test data will be larger.

**Runtime:** Our method requires 2 likelihood estimations and several optimization iterations for each testing image, which will make it slower than its competing methods. As a comparison, input complexity adjusted likelihood does not have computationally overhead, and likelihood ratio with background model needs to train 2 models and take 2 likelihood estimations. However, we observe that in our experiments, on average the computation of LR takes less than 0.3s, which is comparable to the IWAE likelihood estimation (also around 0.3s), so the computational overhead is acceptable.

**Additional Results:** In Appendix C, we include OOD results of models trained on MNIST and SVHN for completeness. We observe that our method works well in these experiments, while competing OOD scores still exhibit severe issues. In appendix E, we show some examples of reconstruction before and after the optimization of encoder. In Appendix F, we show more qualitative examples of reconstruction on different test datasets. In Appendix H, we display some randomly generated samples from the VAEs.

## 6 Conclusion

In this paper, we carefully study the task of unsupervised out-of-distribution detection for VAEs. We evaluate some current state-of-the-art OOD detection scores on a set of experiments, and we conclude that their success on OOD detection for other probabilistic models cannot be easily transferred to VAEs. We also try to provide clues to show that OOD detection is harder for VAEs than for other generative models. To overcome the difficulty, we propose Likelihood Regret, an OOD score for VAEs that is effective on all the tasks we evaluated. We believe that Likelihood Regret can be extended to other generative models if a good optimizable model configuration is defined. We hope this work can lead to further progress on OOD detection for probabilistic generative models.

## Broader Impact

Out-of-distribution detection is a research direction with significant social impact. Nowadays, deep learning is deployed on many systems that are making critical decisions, such as medical diagnosis, factory manufacturing, autonomous driving. Being able to detect anomalous cases is crucial for these systems. Therefore, our work, together with many other research efforts on out-of-distribution detection, is very important for the future development of artificial intelligence. However, we should be cautious on solely relying on algorithmic anomaly detection, as there is always the risk of certain anomalous cases that can fool the algorithms. It is very risky to completely trust these imperfect algorithms.

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
