[Supplementary Material]

# A Experimental Settings

In this section, we introduce detailed settings of our experiments.

## A.1 Datasets

We use several publicly available datasets in our experiments. These datasets include MNIST [25], Fashion-MNIST[47], KMNIST[8], notMNIST, CIFAR-10 and CIFAR-100[23], SVHN [36], CelebA [31] and LSUN[50]. We also create two types of synthetic images: Noise and Constant. Noise images are random samples from Unif$\{0, ..., 255\}$ distribution for each pixel, and Constant images are images with constant value sampled from Unif$\{0, ..., 255\}$ for each channel. Some examples of our created images are shown in Figure 5 We also present some examples of KMNIST and NotMNIST, as they are less familiar to the public. They are used as OOD datasets for models trained on Fashion MNIST.

(a) Noise

(b) Constant

(c) KMNIST

(d) NotMNIST

Figure 5: Some examples of our created images used in experiments.

For most of our experiments, the VAE is trained on Fashion-MNIST and CIFAR-10, where we use the training partition of the datasets. For other datasets used for testing, we use a test partition if it is available, and use randomly sampled data if no predefined partition is available.

| Encoder | Decoder |
|---|---|
| Input $x$ | Input $z$, reshape to nz $\times 1 \times 1$ |
| $4 \times 4$ Conv$_{\text{nf}}$ Stride 2, BN, ReLU | $4 \times 4$ Deconv$_{4 \times \text{nf}}$ Stride 1, BN, ReLU |
| $4 \times 4$ Conv$_{2 \times \text{nf}}$ Stride 2, BN, ReLU | $4 \times 4$ Deconv$_{2 \times \text{nf}}$ Stride 2, BN, ReLU |
| $4 \times 4$ Conv$_{4 \times \text{nf}}$ Stride 2, BN, ReLU | $4 \times 4$ Deconv$_{\text{nf}}$ Stride 2, BN, ReLU |
| $4 \times 4$ Conv$_{2 \times \text{nz}}$ Stride 1 | $4 \times 4$ Conv$_{256 \times \text{nc}}$ Stride 2 |

Table 5: Network structure for VAE based on DCGAN. nz $= 100$ for all models. For VAE trained on Fashion MNIST, nf $= 32$, nc $= 1$; for VAE trained on CIFAR-10, nf $= 64$, nc $= 3$.

We resize images to spatial dimension $32 \times 32$ for all datasets. We trained VAE on Fashion MNIST using $32 \times 32 \times 1$ images, and when we test it on color images, we use only the first channel of the color image. When we use Fashion MNIST or MNIST images to test the VAE trained on CIFAR, we copy the channel three times to make them $32 \times 32 \times 3$.

When computing quantitative metrics, we use 5000 randomly chosen images from in-distribution and OOD datasets, respectively.

## A.2 Implementation Detail

The training and testing of our models largely follow the setting of [34]. In particular, we train VAEs with the DCGAN [39] structure. We present the network structure in Table 5.

On Fashion MNIST we train the VAE for 100 epochs with constant learning rate $5e - 4$ using Adam optimizer and batch size $64$. On CIFAR-10 we train the VAE for 200 epochs with constant learning rate $5e - 4$ using Adam optimizer and batch size $64$. When computing Likelihood Regret, we have the choice of optimizing the whole encoder or only optimizing the mean and variance of posterior. For the former, we start with the trained encoder and optimize its parameters for 100 steps using the Adam optimizer with learning rate $1e - 4$. For the later, we start with the encoding mean and variance, and run optimization for 300 steps using Adam optimizer with learning rate $1e - 4$.

## A.3 Implementing Competing Methods

Input complexity adjusted likelihood [43] is computed by subtracting a measure of the input's complexity from the negative log likelihood with. The input complexity can be obtained from the length of the binary string returned by some lossless compression algorithms. We simply follow their work and use PNG compression and JPEG2000 compression implemented in OpenCV.

Likelihood ratio with background model [40] is computed by subtracting the log likelihood of the background model from the log likelihood of the main model. The background model is trained by perturbing a proportion of randomly chosen pixels, where the perturbation is done by replacing the pixel value by a uniformly sampled random value between 0 and 255. One of the key hyper-parameters $\mu$, is the percentage of pixels to be perturbed. The authors suggest to choose $\mu$ between $0.1$ and $0.3$. We do a simple grid search on $\{0.1, 0.2, 0.3\}$, and use the best one ($0.3$ for Fashion MNIST, and $0.2$ for CIFAR-10). We trained the background VAE with the same setting of the main VAE, except that we apply $\lambda = 10$ $L_2$ weight decay as suggested by the authors.

Latent Mahalanobis distance [4] combines the reconstruction loss and Mahalanobis distance in the latent space as an OOD score. In particular, their score is defined by

$$\text{novelty}(\mathbf{x}) = \alpha \cdot D_M(E(\mathbf{x})) + \beta \cdot \ell(x, D(E(\mathbf{x}))),$$

where $D$ and $E$ are the decoder and encoder, respectively. The second term is the reconstruction loss, and the first term is the Mahalanobis distance ($D_M$) between the encoded sample and the mean vector of the encoded training set. Their method is designed for auto-encoders, so we take $E(x)$ to be the mean of the variational posterior output by the encoder. As suggested by the author, to balance the two terms, $\alpha$ was set to the reciprocal of the standard deviation of the Mahalanobis distance between the encoded validation data and the mean latent train vector, and $\beta$ to the reciprocal of the standard deviation of the reconstruction error on the validation set.

# B Additional Quantitative results: AUPRC and FPR80

|  | $LR_E$ | $LR_Z$ | Likelihood | IC (png) | IC (jp2) | Likelihood Ratio | LMD |
|---|---|---|---|---|---|---|---|
| MNIST | **0.980** | 0.938 | 0.344 | 0.923 | 0.563 | 0.917 | 0.866 |
| CIFAR-10 | 0.995 | 0.998 | **1** | 0.904 | **1** | 0.912 | 0.998 |
| SVHN | 0.998 | **1** | 0.996 | 0.993 | 1 | 0.621 | 0.999 |
| KMNIST | **0.993** | 0.985 | 0.734 | 0.642 | 0.568 | 0.984 | 0.962 |
| NotMNIST | 0.999 | **1** | 0.935 | 0.943 | 0.953 | 0.996 | 0.999 |
| Noise | 0.983 | 0.954 | **1** | 0.4342 | **1** | **1** | **1** |
| Constant | 0.999 | **1** | 0.915 | **1** | **1** | 0.628 | 0.985 |

(a) VAE trained on Fashion MNIST

|  | $LR_E$ | $LR_Z$ | Likelihood | IC (png) | IC (jp2) | Likelihood Ratio | LMD |
|---|---|---|---|---|---|---|---|
| MNIST | 0.993 | 0.979 | 0.307 | **0.997** | 0.988 | 0.665 | 0.308 |
| FMNIST | 0.980 | 0.956 | 0.314 | 0.982 | **0.987** | 0.656 | 0.337 |
| SVHN | 0.841 | 0.799 | 0.343 | **0.922** | 0.914 | 0.337 | 0.389 |
| LSUN | **0.681** | 0.610 | 0.508 | 0.611 | 0.408 | 0.611 | 0.520 |
| CelebA | **0.715** | 0.634 | 0.474 | 0.572 | 0.509 | 0.405 | 0.524 |
| Noise | 0.940 | 0.814 | **1** | 0.313 | 0.318 | **1** | 0.964 |
| Constant | 0.965 | 0.924 | 0.445 | **1** | **1** | 0.410 | 0.593 |

(b) VAE trained on CIFAR-10

Table 6: AUCPRC of Likelihood Regret (LR) and other OOD detection scores on different datasets.

|  | $LR_E$ | $LR_Z$ | Likelihood | IC (png) | IC (jp2) | Likelihood Ratio | LMD |
|---|---|---|---|---|---|---|---|
| MNIST | 0.02 | 0.09 | 0.97 | 0.1 | 0.68 | 0.15 | 0.2 |
| CIFAR-10 | 0 | 0 | 0 | 0.16 | 0 | 0.02 | 0 |
| SVHN | 0 | 0 | 0 | 0 | 0 | 0.35 | 0 |
| KMNIST | 0.01 | 0.01 | 0.51 | 0.44 | 0.57 | 0.02 | 0.06 |
| NotMNIST | 0 | 0 | 0.01 | 0.11 | 0.01 | 0.01 | 0 |
| Noise | 0 | 0.06 | 0 | 0.57 | 0 | 0 | 0 |
| Constant | 0 | 0 | 0.06 | 0 | 0 | 0.27 | 0.01 |

(a) VAE trained on Fashion MNIST

|  | $LR_E$ | $LR_Z$ | Likelihood | IC (png) | IC (jp2) | Likelihood Ratio | LMD |
|---|---|---|---|---|---|---|---|
| MNIST | 0.01 | 0.02 | 1 | 0.01 | 0 | 0.27 | 1 |
| FMNIST | 0.02 | 0.06 | 0.99 | 0.01 | 0.01 | 0.32 | 0.98 |
| SVHN | 0.18 | 0.27 | 0.97 | 0.05 | 0.1 | 0.91 | 0.97 |
| LSUN | 0.46 | 0.55 | 0.77 | 0.5 | 0.86 | 0.62 | 0.77 |
| CelebA | 0.37 | 0.48 | 0.68 | 0.55 | 0.67 | 0.82 | 0.67 |
| Noise | 0.02 | 0.08 | 0 | 0.99 | 0.93 | 0 | 0 |
| Constant | 0.03 | 0.09 | 0.98 | 0 | 0 | 0.65 | 0.96 |

(b) VAE trained on CIFAR-10

Table 7: FPR80 of Likelihood Regret (LR) and other OOD detection scores on different datasets.

# C Results of Model Trained on SVHN

In Table 8, we include results of a set of simple experiments for VAE trained on SVHN. Note that in this case, likelihood itself works well on most tasks. We do not show results of VAE trained on MNIST because every method works nearly perfectly. We observe that LR achieves good performances on all the tasks, while input complexity still has trouble with distinguishing noise from in-distribution data. In addition, its performance on SVHN v.s. CIFAR-10 lies far behind LR. As in Table 2, likelihood

ratio has trouble on CIFAR-10 v.s. Constant. These failures suggest that competing OOD scores have systematical issues on VAE.

|          | $LR_E$ | $LR_Z$ | Likelihood | IC (png) | IC (jp2) | Likelihood Ratio | LMD   |
|----------|--------|--------|------------|----------|----------|------------------|-------|
| MNIST    | **1**  | **1**  | **1**      | **1**    | **1**    | **1**            | **1** |
| FMNIST   | **1**  | 0.999  | **1**      | **1**    | **1**    | 0.928            | 0.998 |
| CIFAR-10 | 0.924  | 0.842  | **0.982**  | 0.524    | 0.608    | 0.936            | 0.955 |
| Noise    | **1**  | **1**  | **1**      | 0.235    | 0.105    | **1**            | **1** |
| Constant | **1**  | **1**  | 0.213      | **1**    | **1**    | 0.136            | 0.818 |

Table 8: AUCROC for model trained on SVHN

## D  Issues with Input Complexity on Glow

In this section, we show the OOD results of input complexity adjusted likelihood for Glow trained on SVHN and MNIST. Note that this set of experiments are not performed by the authors of [43]. We use the same Glow structure used in [43], and train Glow on SVHN and MNIST dataset. We use input complexity adjusted likelihood to detect OOD samples from Fashion MNIST and CIFAR-10. Basically this is the reverse of the commonly conducted experiments. The AUCROC of MNIST v.s. Fashion MNIST is 0.633, and the AUCROC of SVHN v.s. CIFAR-10 is 0.518. Both results suggest that input complexity adjusted likelihood may not work in general, even on flow based model.

## E  Illustration of Optimization

To better illustrate Likelihood Regret, we compare the reconstruction of some test samples with trained VAE and optimized posterior distribution in Figure 6. Since as we show in Figure 2, the VAE can reconstruct both the in-distribution and out of distribution data very well, visually we cannot see obvious differences between reconstructed images from VAE and from optimized $q_\phi(\mathbf{z}|\mathbf{x})$. However, we can still observe the improvements of reconstruction by the optimal $q_\phi(\mathbf{z}|\mathbf{x})$ in MNIST examples.

(a) Fashion MNIST

(b) MNIST

(c) CIFAR-10

(d) SVHN

Figure 6: For each subfigure, the top row contains original images, the middle row contains the reconstruction from VAE, and the bottom row contains the reconstruction images with optimized encoder. **(a), (b)** are obtained from VAE trained on Fsahion MNIST. **(c), (d)** are obtained from VAE trained on CIFAR-10.

## F  More Reconstruction Examples

In this section, we present examples of reconstructed images in different datasets. See Figure 7 and Figure 8. We observe that VAE trained on CIFAR-10 can reconstruct images from multiple datasets very well.

(a) Fashion MNIST

(b) MNIST

(c) CIFAR-10

(d) SVHN

Figure 7: Some examples of reconstructed images using VAE trained on Fashion MNIST. For each subfigure, the first 5 rows are original images and the last 5 rows are their corresponding reconstructions.

# G   Robustness of LR w.r.t the Capacity of VAEs

We present results on VAEs with different capacity in Table 9.

# H   Randomly Generated Samples

We show some randomly generated samples from VAEs in Figure 9. Although the samples are blurry and a bit noisy (due to the cross entropy loss), the semantics of the samples suggest that our VAEs model the in-distribution data well.

(a) Fashion MNIST

(b) MNIST

(c) CIFAR-10

(d) SVHN

Figure 8: Some examples of reconstructed images using VAE trained on CIFAR-10. For each subfigure, the first 5 rows are original images and the last 5 rows are their corresponding reconstructions.

|  | $C = \frac{1}{4}\times$ | | $C = \frac{1}{2}\times$ | | $C = 2\times$ | | $C = 4\times$ | |
|---|---|---|---|---|---|---|---|---|
|  | Likelihood | $\text{LR}_\text{E}$ | Likelihood | $\text{LR}_\text{E}$ | Likelihood | $\text{LR}_\text{E}$ | Likelihood | $\text{LR}_\text{E}$ |
| MNIST | 0.110 | 0.966 | 0.148 | 0.986 | 0.125 | 0.975 | 0.237 | 0.984 |
| KMNIST | 0.608 | 0.991 | 0.65 | 0.996 | 0.711 | 0.992 | 0.812 | 0.994 |
| NotMNIST | 0.977 | 1 | 0.978 | 1 | 0.99 | 0.999 | 1 | 0.997 |
| Noise | 1 | 0.987 | 1 | 0.999 | 1 | 0.997 | 1 | 1 |
| Constant | 0.966 | 0.998 | 0.947 | 0.998 | 0.957 | 0.994 | 0.979 | 0.997 |

(a) $\beta-$VAEs trained on Fashion MNIST

|  | $C = \frac{1}{4}\times$ | | $C = \frac{1}{2}\times$ | | $C = 2\times$ | | $C = 4\times$ | |
|---|---|---|---|---|---|---|---|---|
|  | Likelihood | $\text{LR}_\text{E}$ | Likelihood | $\text{LR}_\text{E}$ | Likelihood | $\text{LR}_\text{E}$ | Likelihood | $\text{LR}_\text{E}$ |
| SVHN | 0.198 | 0.868 | 0.176 | 0.875 | 0.203 | 0.814 | 0.198 | 0.79 |
| LSUN | 0.462 | 0.641 | 0.453 | 0.652 | 0.445 | 0.635 | 0.467 | 0.611 |
| CelebA | 0.455 | 0.706 | 0.445 | 0.717 | 0.521 | 0.793 | 0.475 | 0.753 |
| Noise | 1 | 1 | 1 | 1 | 1 | 0.998 | 1 | 0.999 |
| Constant | 0.218 | 0.999 | 0.276 | 0.998 | 0.292 | 0.989 | 0.246 | 0.962 |

(b) $\beta-$VAEs trained on CIFAR-10

Table 9: AUCROC of Likelihood Regret (LR) and Likelihoods for VAEs with different capacity, where we proportionally increase/decrease the number of channels of convolution layers. For example, $C = \frac{1}{4}\times$ means that the VAE has $\frac{1}{4}$ channels compared to the baseline VAE.

(a) Fashion MNIST  (b) CIFAR

Figure 9: Some examples of randomly generated samples.