[Reviews · NeurIPS 2020]

Review 1

Summary and Contributions: This paper proposes a new out-of-domain (OOD) detection approach for Variational Autoencoders (VAEs). The authors explain that BDP computed for VAEs changes by a smaller amount compared to Flows and PixelCNNs, and this is the reason the previously proposed OOD scores don’t work well for VAEs. The proposed score is based on the improvement of likelihood on new examples if the encoder is further optimized on the new example.

Strengths: As the authors point out, deep generative models (and in particular VAEs) fail to assign lower p(x) to OOD samples, and the proposed OOD scores so far do not seem to address this. While one line work involves coming up with better models and objectives, another reasonable approach is what this paper focused on which is designing better metrics. And the proposed approach makes sense and seems to address this problem to some extent based on the experiments.

Weaknesses: I had a question for the authors. Something I often wonder about OOD scores is how good are they at differentiating between “difficult” in-domain and OOD examples. For example, if the model is trained on (red squares, red circles, blue squares), can the proposed score detect that a ‘blue circle’ is in-domain but say a ‘car’ image is OOD? Also I was wondering if the authors have done any architecture search? Working on VAEs myself I know that deep and shallow VAE generalization differ vastly.

Correctness: Yes

Clarity: This is very well-written paper. I found it really easy to read overall. I have a small complaint about the notation. It is common in the VAE literature for θ to be the parameters of the generative model. In the paper, the generative model is defined as pθ(x), but then define the parameters of the decoder as: pη(x|z) and define θ as (η,φ) which doesn’t make sense given that p(x) theoretically doesn’t depend on the encoder.

Relation to Prior Work: I am not very familiar with SOTA models for OOD detection but the authors seem to cover the literature review well.

Reproducibility: Yes

Additional Feedback: Typically people use a linear layer after the last Conv layer in the encoder. I noticed that in Table 3 that this is not the case. I was curious if this makes a difference.


Review 2

Summary and Contributions: This paper developed an elegant yet effective method to detect OOD examples using VAE, which was termed Likelihood Regret (LR). LR is obtained by taking the difference between the log likelihood under the original VAE model trained on the entire training set and the log-likelihood of the same sample but “fine-tuned” on this specific sample -- either by finding the optimal posterior for that single sample or optimizing the encoder only for that single sample. They showed that the LR method is superior to the likelihood baseline and previous methods (including input complexity adjusted score, likelihood ratio , latent Mahalanobis distance) on VAE, where previous methods have some failure modes on some OOD datasets.

Strengths: The method is simple and elegant yet very powerful for OOD detection. It's a really interesting and well-written paper. There is very little computational overhead for this method, on the scale of seconds, which makes it very practical to actually be used. The problem statement is very well-motivated and communicated clearly, the authors made it very clear what is missing: an OOD detection sore for VAE, in Sections 1 and 2.2 and 3. Nice analysis on lines 249-259 on how the complexity measurement gap can override the likelihood gap. Experimentation is complete and comprehensive, for VAEs. Authors compared the proposed method with prior methods including likelihood ratio and input complexity adjusted score, the likelihood baseline. And even better two implementations of the proposed method are compared.

Weaknesses: My main concern is on the lack of comparison with previous methods (input complexity adjusted score and the likelihood ratio method) on the corresponding models(e.g. Glow and Pixel-CNN) in the original papers instead of just comparing their methods on VAE. Those methods were developed and tested with the corresponding generative models in the original paper and it seems unfair to only compare with their method on VAEs. Without this comparison, if a researcher wants to choose the SOTA OOD detection method for their own applications, it’s hard to tell which method will most likely achieve the best performance if they have the freedom to choose their own generative models. This is the main drawback and the main reason for my rating. Furthermore, this leads to the general motivation of the paper. I really like the analysis on why prior likelihood-ratio based methods didn’t work as well on VAEs, however, if all we care about is detecting OOD examples, why is it absolutely necessary to have a method that works well on VAEs? It’d be great if the authors could explain and motivate the necessity. Additionally, it’d be very helpful to include an analysis, or hypothesis, on why optimizing the encoder leads to slightly better performance than optimizing the posteriors only. Minor concerns: I assume the y axis of the histograms in Figure 1 is frequency, it’d be great if you could add y axis along with the unit for clarity. Same with Figure 3. Since the “tao” notation is central to this paper, It’d be helpful to spend a sentence to explain, in English, what this is and how to interpret it. When describing the experiments from lines 214-220, clarify that the previous methods were tested for VAE instead of the generative models in the original papers. It’d be helpful to expand and explain more on how the bottleneck structure of VAE provides a natural regularization, in lines 154-156. Updates: I have read the rebuttal and the other reviews and based on the new results I have decided to increase my score from 6 to 7. I was not trying to say it's not useful to have a method on OOD detection using VAEs, but rather the point is that I think when introducing a new method, it's important to provide thorough empirical evidence and analysis on all major types of deep generative models, it's ok if this method works better on one model than the rest, but it's important to provide complete information for future research. I really appreciate the follow up experiments, however it still seems a bit rushed as the results were achieved by fine tuning only the last layer to prevent overfitting and there are other alternatives that are not tested on other generative models, such as optimizing the latent code directly. This rush is very understandable under such limited time of 1 week. However I do think it's important to do more thorough experiments of this sort to really make sure not to misguide the community.

Correctness: Yes. Authors claimed that their LR method obtains the best overall OOD detection performances when applied on VAE and their detailed experimentation indeed verified and support this claim by testing their method on 7 datasets and comparing with 3 methods in prior work along with a likelihood baseline, and the results indeed indicated the superiority of their LR method, on VAE.

Clarity: The paper is very well-written and clear and easy to read. Introduction is comprehensive and clear by pointing out the failure of current OOD scores on VAE presents a gap in current research literature and that their method can fill this gap. Nice summary of results in 5.2. It’ll probably be worth repeating throughout the paper that everything tested in this paper is on VAEs, as sometimes the prior methods were on other generative models and it might confuse readers who are only skimming the paper.

Relation to Prior Work: Yes, the major prior works were discussed along with their limitations, and how the author's LR method can overcome those limitations.

Reproducibility: Yes

Additional Feedback: In the related work section near the end of paragraph two where comparing a batch of samples were discussed, maybe it’d also be helpful to consider mentioning a comprehensive study on detecting dataset shift: https://papers.nips.cc/paper/8420-failing-loudly-an-empirical-study-of-methods-for-detecting-dataset-shift.pdf Line 51: A slightly more detailed summary of experimental results will give the reader a better overall idea of the paper, e.g, “obtains the best overall performances”... compared to what?


Review 3

Summary and Contributions: The paper proposes a way to do out-of-distribution detection with a given variational auto-encoder. For this, they define Likelihood Regret as the difference in ELBO made by replacing the VAE's encoding (or alternatively encoder) with an optimal one for the given sample. Using this number for discriminating in- and out-distribution, they generally improve OOD detection performance compared to other VAE-based methods and appear to not have unacceptable weaknesses on any of the standard evaluation dataset pairs, which all compared previous methods have.

Strengths: Having good OOD detection properties is desirable for different kinds of machine learning models that by themselves do not need to be focused on solving that task. In that sense, it is useful to enable the popular VAE's being used in that regard, especially since the method does not need any adjustments on the model itself (training, architecture) and therefore does not impact its performance in other regards. The paper gives a good theoretical and methodological derivation of the proposed Likelihood Regret without needing any difficult to access assumptions or hypotheses. The applicable algorithm follows directly from these theoretical considerations. The results, while not being remarkable for general OOD detection methods, are very good when compared to other VAE based methods, especially since they do not show catastrophic failure on any of the presented out-distributions (which include a good standard range of datasets).

Weaknesses: The results are only good for the quite restricted case of using a VAE for OOD detection. It is not clear why this is an important use case, since e.g. classifier based OOD detection seems to work better than generative model / density / likelihood based approaches. Non-VAE types of the latter class, as stated, also work better in that regard. As several modifications of the VAE (beta-VAE etc.) have been proposed and become popular, a transfer to them should be discussed and if possible evaluated. If there are more points for why using particularly VAEs for OOD detection, their inclusion would be welcome. Even if all methods work nearly perfectly well, the results on MNIST should be included in the supplement, ideally with the inclusion of the EMNIST letter dataset where other OOD detection method do not achieve perfect AUCROC yet. For the CIFAR-10 in-distribution, the CIFAR-100 dataset should be included as out-distribution since it represents very similarly captured images and is the most challenging one for many approaches. The AUCROC of CIFAR-10 vs. SVHN is not close to the optimal value 1, since 87% still represents quite many confused pairs.

Correctness: The mathematical derivations are clear and seem to be correct. The demo code contains typos in lines 90 and 131 and then gives a FileNotFoundError after the evaluations on the in-distribution are done (their number should be set lower than 999 if it is just about printouts of the ELBOs). The dataloaders should probably have shuffle=False to get reproducible evaluation runs.

Clarity: Yes, it is very well written, set and structured. Minor remarks: Please check for consistent spelling of dataset names and "vs.".

Relation to Prior Work: Yes, prior work and other approaches are referenced.

Reproducibility: Yes

Additional Feedback: I couldn't find the density-based definition of OOD in [15]. Could you please point out where they make it? Splitting off the presentation of the FashionMNIST vs. MNIST and CIFAR-10 vs. SVHN results from the rest seems unnecessary. For comparison, AUCROC values of non-VAE based methods could be included in the appendix. Update: Thanks for the new points on the relevance of VAEs and generative models in general for OOD detection and for the additional evaluations. I've decided to raise my score from 6 to 7. I think the method is an interesting improvement for the limited scope that it is proposed for and the paper is well written. Thus it might be a small but valuable step towards understanding the behaviour of likelihood estimations on unseen data. My concerns have been well addressed in the rebuttal, however the CIFAR-10 vs. CIFAR-100 score is a bit disappointing, especially as classifier based methods achieve around 90% AUCROC; since the reported 58.2% improves above other likelihood-based methods, this is ok though. The MNIST-EMNIST score is quite impressive on the other hand. The first rebuttal point about generative models working with unlabelled data in contrast to classifier based OOD detection for me adds some convincing justification for the relevance of using VAEs here. An extended discussion of this and the other items should be included in the potential final version. From my point of view, the improvement on VAEs has merit by itself, but a comprehensive treatment of all commonly used generative models (more in-depth than the first results shown in the rebuttal) would be very beneficial.


Review 4

Summary and Contributions: The paper tackles the problem of OOD detection via generative models. It highlights how current OOD scores perform poorly when applied in the context of VAEs, it proposes a new OOD score - likelihood regret score - and it shows that it generally outperforms alternatives. In particular, while the alternatives perform poorly on at least one setting, the proposed method is consistently among the top-performing methods.

Strengths: * The OOD detection problem is important to the community and recent works have attempted to provide OOD scores that outperform log-likelihood thresholding at OOD detection. From this perspective, the current paper is relevant to the community. * The paper illustrates clearly that likelihood thresholding can fail to detect OOD samples in the VAE case and demonstrates that the proposed approach can alleviate this issue. * The experimental results are strong, with the proposed approach outperforming the generic OOD scores. * The paper makes a reasonable attempt to explain why the VAEs might be different than other generative models when it comes to OOD detection. * The paper also discusses its limitations (e.g. the increased computational time).

Weaknesses: I generally like this investigation. My main concerns are the following: * Previous work has already established that likelihood thresholding can fail to detect OOD samples and proposed OOD scores to mitigate this issue. The current approach introduces a VAE-specific score. From this perspective, the current approach may have a limited impact (that has to do with how often is a VAE the generative model of choice for performing OOD detection). * Related to the above, the paper would have a stronger impact evaluated VAE-based OOD detection with alternative generative models on the same datasets, and it highlighted that a VAE can be necessary and therefore a VAE-specific OOD score is necessary as well. * Given the restriction to VAEs, another question is whether the results hold only for image VAEs, or would also hold for other domains (e.g. VAEs have been used to model sequences and graphs).

Correctness: * The proposed approach is sensible. * The main claims, i.e. (1) current OOD scores do not perform well for VAE and (2) the likelihood regret score is effective across tasks, are supported well by evidence.

Clarity: The paper has been very easy to follow.

Relation to Prior Work: The paper mentioned the relevant prior work that I'm familiar with. The paper distinguishes itself by proposing a VAE-specific approach.

Reproducibility: Yes

Additional Feedback:

[Author Response · NeurIPS 2020]

We thank all reviewers for their useful comments. First we address the common concern regarding the **importance and impact of our work** in the following three arguments:

1. Although classifier based OOD detectors often outperform OOD detectors based on generative models, we should note that many machine learning systems deployed in practice are trained on unlabeled data, and generative models are widely used in tasks like language modeling and image editing. Therefore, OOD detection with unsupervised generative models is an important task.

2. Among likelihood based generative models, VAEs have the promising property of extracting representations from data, and they are particularly popular in applications (for example, VAE is a component of most image-to-image translation frameworks). Recent work obtains impressive OOD detection on flow/auto-regressive models, but our motivation is that, if VAE is being used in a task, it would be beneficial to enable the VAE itself to detect OOD inputs, rather than training a separate generative model merely for OOD detection. However, as we show, recent SOTA OOD scores for generative models cannot be applied to VAEs. Our work makes an important contribution by introducing an easy yet effective OOD score that works well on VAEs.

3. It is most natural to apply LR to VAEs due to the bottleneck structure serving as a regularization for optimizable model configuration, however, later we also explore the possibility of applying LR to flow/pixelCNN. In Table 1, we report the AUCROC on some tasks obtained from LR on Glow and PixelCNN. We only optimize the last coupling block and the last masked conv layer for Glow and PixelCNN respectively to avoid overfitting on a single input. Our results are competitive to those of previous SOTA methods on generative models. We therefore highlight LR as a general approach for OOD detection of all likelihood based generative models, with unique effectiveness on VAEs.

|          | Glow  | PixelCNN |
|----------|-------|----------|
| MNIST    | 0.995 | 0.987    |
| KMNIST   | 0.990 | 0.973    |
| NotMNIST | 0.991 | 0.995    |
| Noise    | 0.989 | 0.964    |
| Constant | 1     | 1        |

(a) Models trained on Fashion MNIST

|          | Glow  | PixelCNN |
|----------|-------|----------|
| SVHN     | 0.894 | 0.889    |
| CelebA   | 0.735 | 0.774    |
| LSUN     | 0.643 | 0.686    |
| Noise    | 1     | 0.975    |
| Constant | 0.989 | 1        |

(b) Models trained on CIFAR-10

Table 1: AUCROC obtained from Likelihood Regret on Glow and PixelCNN.

We now address detailed concerns of each reviewer.

**Reviewer 1:** We apologize for the confusion caused by the notation. Indeed, the decoder should be denoted with $\theta$. Our VAE is based on the DCGAN structure, which is fully convolutional and the last FC layer is replaced by conv layer that reduces the spatial dimension to $1 \times 1$ (which returns the vectors for posterior mean/variance). We tried different network structures, including those with FC layers, and have found little difference in terms of OOD detection. The precise definition of OOD samples is interesting and debatable, as we only have an in-distribution *dataset* rather than a true distribution. Strictly speaking, both blue circle and cars are OOD, but the former is much harder to detect, especially for generative models.

**Reviewer 2:** Please refer to items 1-3 for concerns regarding why we focus on VAE's OOD detection. In short, our motivation is to let VAEs detect OOD when it is used, and our extended experiments on flow/pixelCNN indicate that LR is general across generative models. As for why optimizing the encoder works better, for in-dist samples, optimizing the encoder is more constrained than directly optimizing $z$, which prevents the latent variables from moving too much. We will provided a detailed explanation in the revision. We will carefully fix writting issues as pointed out.

**Reviewer 3:** Please refer to items 1-3 for concerns regarding why we focus on VAE's OOD detection. We found that $\beta$-VAEs behave similarly to plain VAEs in terms of OOD detection. We will include more details on applying our method to other VAE variants. We will also include the MNIST results in the appendix as suggested. In particular, we obtain AUCROC 0.963 on your mentioned MNIST vs. EMNIST (VAE's likelihood gives 0.781). CIFAR-10 vs. CIFAR-100 is indeed a very hard task for generative models. We tested LR on this task and obtained AUCROC 0.582 (compared to 0.489 using likelihood), while likelihood ratio gives 0.564 and IC gives 0.535. This result is also comparable to IC on pixelCNN. We thank the reviewer for pointing out writing issues and typos in the code. These will be fixed, suggested additions will be included, as well as the correct reference for the definition of OOD.

**Reviewer 4:** Please refer to items 1-3 for concerns regarding why we focus on VAE's OOD detection and applying LR to other generative models. We only tested on image data because most recent OOD papers, both classifier based and generative model based, focus on image data. The Likelihood Ratio paper by Ren et al. introduces a gene sequence OOD dataset, and we tested our Likelihood Regret on a VAE trained on this dataset. Our LR obtains 0.745 AUCROC (the VAE's likelihood baseline gives 0.538). This result is competitive to Ren et al. (which uses a auto-regressive model whose likelihood gives 0.626 and their proposed likelihood ratio improves to 0.732).

[Meta-Review · NeurIPS 2020]

The key idea of this paper is checking how well a given VAE can be further trained on a given test input. The hope is that training the encoder for further iterations may increase the likelihood for OOD samples more when compared to inliers to facilitate detection. The authors characterize this improvement by a measure, coined as likelihood regret. The authors do not provide any analysis why this method might work, or characterize the conditions when it might not. This is not a requirement, but then the paper should provide enough empirical evidence that the approach is noteworthy. Overall, the paper has been perceived positively, and the authors have provided additional experimental results during the rebuttal. One reviewer found the empirical evaluations not sufficient or hastily executed to be convincing. They noted that the paper may benefit an analysis on all major types of deep generative models.